# Physical properties of lactic acid bacteria influence the level of protection against influenza infection in mice

Takumi Watanabe[1,2]*, Kyoko Hayashi[2], Isao Takahashi[3], Makoto Ohwaki[4], Tatsuhiko Kan[1], Toshio Kawahara[5]*

1 Bio-Lab Co., Ltd., Hidaka, Saitama, Japan, 2 Graduate School of Engineering, Chubu University, Kasugai, Japan, 3 ICAM Co., Ltd., Itabashi, Tokyo, Japan, 4 Non-Profit Organisation, The Japanese Association of Clinical Research on Supplements, Hidaka, Saitama, Japan, 5 College of Life and Health Sciences, Chubu University, Kasugai, Japan

* t.watanabe@bio-ken.jp (TW); toshi@isc.chubu.ac.jp (TK)

**Data Availability Statement:** All relevant data are within the manuscript.

**Funding:** The research received support from Bio-Lab Co., Ltd. and ICAM Co., Ltd. The funder

## Abstract

We evaluated whether the water dispersibility of lactic acid bacteria (*Enterococcus faecalis* KH2) affects their efficacy. When cultured lactic acid bacteria are washed, heat-killed, and powdered, adhesion occurs between results in aggregation (non-treated lactic acid bacteria, n-LAB). However, dispersed lactic acid bacteria (d-LAB) with a lower number of aggregates can be prepared by treating them with a high-pressure homogenizer and adding an excipient during powdering. Mice were administered n-LAB or d-LAB Peyer's patches in the small intestine were observed. Following n-LAB administration, a high amount of aggregated bacteria drifting in the intestinal mucosa was observed; meanwhile, d-LAB reached the Peyer's patches and was absorbed into them. Evaluation in a mouse influenza virus infection model showed that d-LAB was more effective than n-LAB in the influenza yield of bronchoalveolar lavage fluids on day 3 post-infection and neutralizing antibody titers of sera and influenza virus-specific immunoglobulin A in the feces on day 14 post-infection. Therefore, the physical properties of lactic acid bacteria affect their efficacy; controlling their water dispersibility can improve their effectiveness.

## Introduction

Lactic acid bacteria (LAB) play an important role in various fermentation processes. In the book "Essais Optimistes [1]," by the Russian microbiologist Metchnikov, he advocated the consumption of yogurt to increase life expectancy. This led to a flurry of research on the health benefits of LAB. In recent years, many health benefits have been reported, including improved gut microbiota [2, 3] and biological defense [4, 5] as well as anti-allergenic [6, 7] and anti-tumorigenic [8, 9] effects. The isolation and cultivation of LAB has rendered its consumption convenient. The active ingredients present in LAB, including lipoteichoic acid [10] and nucleic acid [11], are involved in modulating the immune response. LAB are used in food supplements, beverages, confectionery, cereals, and others. Moreover, they are consumed in

provided support in the form of salaries for authors (Takumi Watanabe, Tatsuhiko Kan, and Isao Takahashi) but had no role in study design, data collection and analysis, decision to publish, or preparation of the manuscript.

**Competing interests:** Takumi Watanabe and Tatsuhiko Kan are employed by Bio-Lab Co., Ltd. Isao Takahashi is employed by ICAM Co., Ltd. None of the authors had a personal or financial conflict of interest. This does not alter the authors' adherence to PLOS ONE policies on sharing data and materials. In addition, all authors involved with the present manuscript declare that they have no competing interests of any kind, including financial and non-financial competing interests.

powdered form. With the growth of the market for LAB, research and development has increased and various kinds of LAB, such as anti-obesity [12] and antiviral [13] variants, have been developed. However, reports evaluating and verifying LAB powder raw materials produced in a factory are scarce. We believe that there is a difference in physical properties, particularly dispersion, between factory-produced LAB powder and laboratory-prepared LAB powder, based on processes such as thermal history and powderization. Recently, bacteria have been reported to be absorbed by the binding of bacterial S-layer protein to uromodulin of M cells in Peyer's patches [14], phagocytosed by antigen-presenting cells such as dendritic cells and macrophages [15, 16], and recognized by pattern-recognition receptors, such as toll-like receptors, nucleotide binding oligomerization domain-like receptors and retinoic acid inducible gene-like receptors, for the production of immune-related substances, such as cytokines [17]. LAB are absorbed by the microfold cell (M cell) that are scattered on the Peyer's patches; however, if LAB aggregate and become larger than M cells, physically absorbing them becomes difficult. When we investigated some LAB products, we observed that LAB were agglomerated, which is common in powdered materials. Therefore, we resuspended the LAB before and after powderization in distilled water and compared their physical properties in terms of particle size distribution to investigate whether the difference difference in powderization had an effect on LAB. The mean particle size of LAB before powderization was smaller than that after powderization and bacteria aggregated after powderization. This may lead to a loss of the beneficial effects of LAB on health. We prepared a non-agglomerating LAB powder by dispersing bacteria in a high-pressure homogenizer and adding dextrin as a vehicle. Thereafter, LAB powders with higher and lower number of aggregates were compared. Water dispersibility was analyzed using a laser diffraction particle size analyzer, uptake from the Peyer's patches of mice was microscopically observed, experiments on IL-12 production using mouse splenocytes were conducted to evaluate immune response [18, 19], and the protective effect of LAB on viral infection [20–24], the main health effect of LAB was compared in a mouse influenza infection model.

## Materials and methods

### Sample preparation and particle size measurement

*Enterococcus faecalis* KH2 (International Patent Organism Depositary, Japan; number NITE P-14444; GenBank Accession number, AB534553) was stored at Bio-Lab Co., Ltd. LAB were aerobically grown overnight at 37°C in MRS broth (Difco, Detroit, MI, USA) and washed with distilled water, followed by centrifugation at $10,000 \times g$ for 3 min. The suspension of bacteria in distilled water [20–30 mg (wet bacteria weight)/mL] was heated at 105°C for 30 min using an autoclave (HV-25ⅡLB; Hirayama Manufacturing Corp., Saitama, Japan). The untreated LAB powder was designated "non-treated LAB" (n-LAB). To increase the water dispersibility of the prepared LAB, the sample was treated using a high-pressure homogenizer (ECONIZER LABO-01; Sanmaru Machinery Co., Ltd. Shizuoka, Japan) at 15 MPa and an equal amount of dextrin (NSD300; San-ei Sucrochemical Co., Ltd. Aichi, Japan) was added. The powdered sample was designated "dispersed LAB" (d-LAB). A spray dryer (ADL311S-A; Yamato Scientific Co., Ltd. Tokyo, Japan) was used for powderization. Each sample was diluted with distilled water to a concentration of 10 mg/mL, and particle size distribution was measured using a laser diffraction particle size analyzer (SALD-2300; Shimadzu Corporation, Kyoto, Japan) to calculate average and median particle sizes.

### State of LAB in mouse Peyer's patches

LAB was diluted with distilled water to 25 mg/mL. Cy3 (Amersham Cy3 Mono-Reactive Dye Pack, GE Healthcare, Chicago, USA) was added to reach a final concentration of 0.6 mg/mL

and incubated for 2 h in the dark. Thereafter, the samples were centrifuged at 3,000 ×*g* for 15 min and washed with phosphate-buffered saline (PBS)(-). Subsequently, PBS(-) was added to the sample for administration. Male specific pathogen-free Slc:ddY mice (6 weeks old, 16–18 g) were obtained from Tokyo Laboratory Animals Science (Tokyo, Japan). A 100-fold diluted sample was injected into the intestines of mice fasted overnight and incubated for 1 h for the loop assay [25, 26]. After the intestines were collected, actin was stained with phalloidin (Alexa Fluor™ 488 Phalloidin, Thermo Fisher Scientific, Waltham, MA, USA) and nuclei were stained with 4',6-diamidino-2-phenylindole (DAPI, Thermo Fisher Scientific, Waltham, MA, USA) for 1 h. The stained small intestine was observed under a fluorescence microscope (OPTI-PHOTO, NIKON, Tokyo, Japan) or confocal laser scanning microscope (LSM 5EXCITER, ZEISS, Jena, Germany).

## IL-12 production by mouse splenocytes

The LAB suspension was added at a final concentration of 1 μg/mL (culture medium, RPMI1640, Wako, Osaka, Japan) to 6 wells per sample in a 96-well cell culture plate, which was seeded with mouse splenocytes collected from BALB/c mice (8 to 9 weeks old) obtained from CLEA Japan (Tokyo, Japan). The mixtures of mouse cells and bacteria were cultured in a humidified 5% $CO_2$ incubator at 37˚C. After incubation for 24 h, the culture supernatants of the mixtures were collected to determine IL-12 concentration using enzyme-linked immuno-sorbent assay (ELISA). The reagents used for ELISA included a primary antibody [purified anti-mouse IL-12 (p70) antibody, BioLegend Inc., San Diego, CA, USA], secondary antibody (Biotin anti-mouse IL-12/IL-23 p40 antibody; BioLegend), blocking reagent (Block Ace Powder, KAC Co., Ltd., Kyoto, Japan), capture antibody (HRP Avidin, BioLegend), substrate (tetramethylbenzidine, Sigma-Aldrich, St. Louis, MO, USA), and standards [Recombinant Mouse IL-12 (p70) (ELISA Std.), BioLegend]; IL-12 levels were measured using sandwich ELISA [27].

## Model of mouse IFV infection

Female, specific pathogen-free BALB/c mice (5–6 weeks old, 16–18 g) were obtained from Japan SLC (Shizuoka, Japan). All experiments were conducted in accordance with the animal experimentation guidelines of Chubu University and approved by the Animal Care Committee of Chubu University (Permission number: 3010057). Mice were fed conventional diet (CE-2, CLEA Japan, Inc., Tokyo, Japan) and water ad libitum and kept in a temperature and humidity-controlled environment with a 12-h light/dark cycle and a room temperature of 22˚C – 24˚C. Anesthesia was administered using three types of anesthetics (Domitor, Nippon Zenyaku Kogyo Co., Ltd., Fukusima, Japan; Dormicum, Astellas Pharma Inc., Tokyo, Japan; Vetorphale, Meiji Seika Pharma Co., Ltd., Tokyo, Japan) to minimize suffering. No side effects of drugs were detected throughout the experiments. Mice were intranasally infected with influenza A virus (A/NWS/33, H1N1 subtype) [28] at $2 \times 10^4$ plaque-forming units /50 μL per mouse (n = 10) on day 0. n-LAB (5 mg/mouse/day) and d-LAB (because dextrin was mixed with LAB in equal amounts, the dose was doubled to 10 mg/mouse/day) were suspended in distilled water. Oseltamivir phosphate (OSL; 0.2 mg/mouse/day) was used as a positive control for antiviral effects and dissolved in distilled water. n-LAB, d-LAB, or OSL was orally administered two times per day, from day 7 before viral inoculation to day 14 after inoculation. The control mice were orally administered with vehicle (distilled water) alone. Because influenza virus (IFV) infection causes a reduction in body weight [29, 30], mice belonging to each treatment group were weighed daily for day 14, beginning on the day of IFV inoculation (designated day 0). After euthanizing the animals via cervical dislocation by a skilled person to avoid causing pain, lung and bronchoalveolar lavage fluid (BALF) samples were collected from each

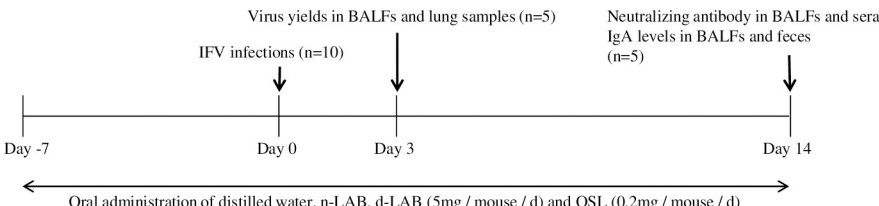

**Fig 1. Experimental procedure of influenza virus infection.** Mice in the control, n-LAB, d-LAB, and OSL groups were administered distilled water, n-LAB (5 mg/day), d-LAB (5 mg/day, two doses daily) and OSL (0.2 mg/day, two doses daily), respectively, during the study period (days −7 to 14). Mice were intranasally infected with IFV on day 0. On day 3 after IFV infection, 5 mice from each group were euthanized to quantify virus load in the lungs and BALF. Further, 5 mice were euthanized for measuring neutralizing antibody and IgA levels on day 14. BALF, bronchoalveolar lavage fluid; d-LAB, dispersed lactic acid bacteria; IFV, influenza A virus; n-LAB, non-treated lactic acid bacteria; OSL, oseltamivir phosphate.

group on days 3 and 14, and blood and fecal samples were collected on day 14 (Fig 1). Lung samples were sonicated for 10 s after the addition of 10 µL PBS per mg of lung tissue and centrifuged at 1,500 rpm for 30 min to separate the supernatants, which were stored at −80˚C. BALF samples were collected after 4 washes with 0.8 mL ice-cold PBS via a tracheal cannula and centrifuged at 1,500 rpm for 10 min; the supernatants were stored at −80˚C. Blood samples were centrifuged at 3,000 rpm for 10 min, and the sera were stored at −20˚C. Fecal extracts were prepared by adding PBS at 10 µL per mg of feces. The amount of virus in the lung and BALF samples collected on day 3 post-infection was quantified using the plaque assays on Madin–Darby canine kidney cell monolayers. Sera and BALF samples were subjected to neutralizing antibody titer assays using the 50% plaque reduction method, as previously described [31, 32]. BALF samples and fecal extracts were assessed for mucosal virus-specific IgA levels by ELISA, as previously described [33].

## Statistical analysis

The effects of the drugs were analyzed using one-way analysis of variance, and correction for multiple comparisons was conducted by Tukey's multiple comparison test. A $p$ value of <0.05 was considered significant.

## Results

### Particle size distribution of n-LAB and d-LAB

Fig 2A shows the particle size distribution of n-LAB that were washed with distilled water and powdered using a spray dryer. Fig 2B shows the particle size distribution of d-LAB that were washed with distilled water, treated with a high-pressure homogenizer, and powdered with an equal amount of dextrin. The mean and median particle sizes of n-LAB and d-LAB are shown in Table 1. The particle sizes of approximately 55 and 0.7 µm were found to be more abundant for n-LAB and d-LAB, respectively size. The mean particle size of d-LAB was smaller than that of n-LAB [0.679 vs. 35.454 µm; the median was 0.633 µm for d-LAB compared with 40.761 µm for n-LAB (Table 1)].

### Observation of mice Peyer's patches after n-LAB and d-LAB administration

Although n-LAB was not visible on Peyer's patches, it was observed on the mucosa around the Peyer's patches (Fig 3A). n-LAB was in an agglomerated state. By contrast, d-LAB was observed on the surface of the Peyer's patches (Fig 3B). In addition, n-LAB showed a larger

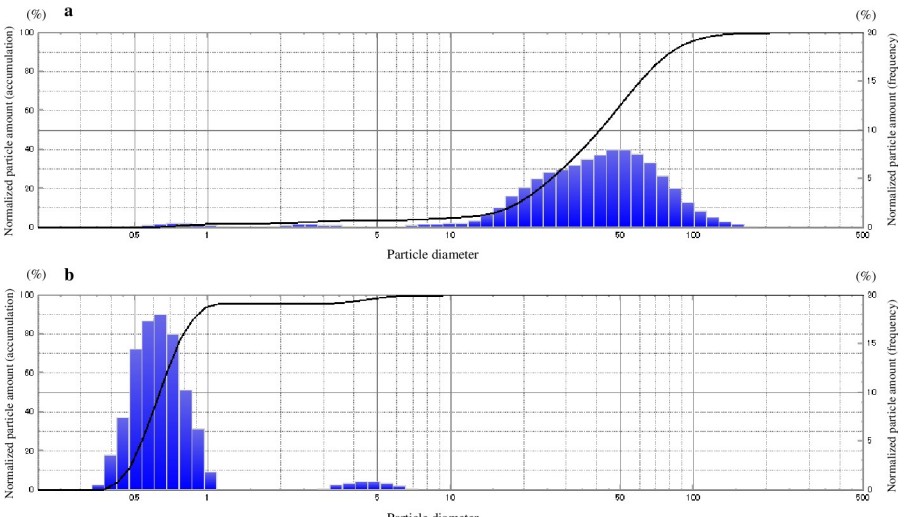

**Fig 2. Measurement of n-LAB and d-LAB using a laser diffraction particle size analyzer (SALD-2300).** n-LAB (**a**) and d-LAB (**b**) were suspended in distilled water and the relative particle mass (frequency and integration) was measured using a laser diffraction particle size analyzer. d-LAB, dispersed lactic acid bacteria; n-LAB, non-treated lactic acid bacteria; n = 3.

bacterial image than d-LAB due to bacterial the aggregation. Furthermore, cLSM was used to confirm the underlying layer of the Peyer's patches upon d-LAB administration and bacterial uptake into the body was present (Fig 3C).

## Effects of n-LAB and d-LAB on IL-12 production in mouse splenocytes

We compared IL-12 production after n-LAB and d-LAB administration using mouse splenocytes and found that d-LAB was significantly higher than n-LAB (Fig 4).

## Effects of n-LAB and d-LAB on IFV infection in mice

The effects of n-LAB and d-LAB on the change in body weight of IFV-infected mice were examined (Fig 5). The control, n-LAB, and d-LAB groups showed a decrease of approximately 15.9%, 14.9%, and 14.4%, respectively, on day 7 post-infection. Although no significant difference was observed between the n-LAB and d-LAB groups, the d-LAB group showed slightly better weight loss suppression compared with the n-LAB group. Thereafter, the mice gradually gained weight; although there was no significant difference, the mice in the d-LAB group reached their pre-infection body weight on day 12 post-infection, whereas those in the control

**Table 1. Measurement of n-LAB and d-LAB using a laser diffraction particle size analyzer.**

| | Particle size (μm) | |
| --- | --- | --- |
| | Mean±SD | Median |
| **n-LAB** | 35.454±0.378 | 40.761 |
| **d-LAB** | 0.679**±0.203 | 0.633** |

n-LAB (**a**) or d-LAB (**b**) were suspended in distilled water and particle sizes (mean and median) were measured using a laser diffraction particle size analyzer (SALD-2300). d-LAB, dispersed lactic acid bacteria; n-LAB, non-treated lactic acid bacteria; n = 3

**p < 0.01 vs. n-LAB.

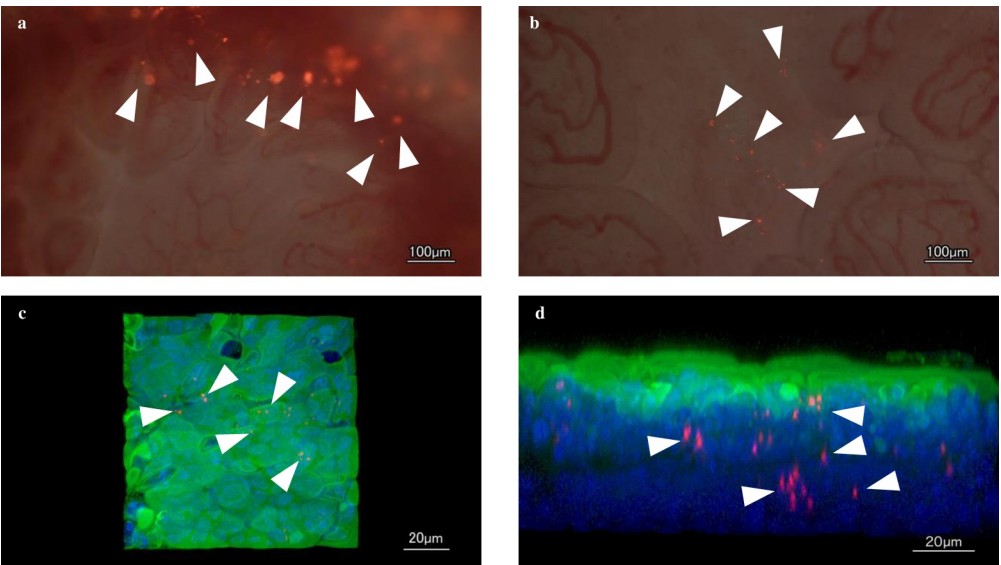

**Fig 3. Microscopic images of the Peyer's patch in mice after LAB and d-LAB administration and Peyer's patches images after d-LAB administration with cLSM.** n-LAB (**a**) or d-LAB (**b**) stained with Cy3 and Peyer's patches were imaged using a fluorescence microscope. The uptake of d-LAB by Peyer's patches after d-LAB administration was imaged using cLSM [(**c**); image from the lumen and (**d**); cross-sectional image]; actin was stained with phalloidin and nuclei were stained with DAPI. The white triangular arrow shows the bacteria. cLSM, confocal laser scanning microscopy; d-LAB, dispersed lactic acid bacteria; n-LAB, non-treated lactic acid bacteria.

and n-LAB groups did not return to their pre-infection body weight even on day 14 post-infection. There were no differences in body weight change in each group during the 7-day administration period.

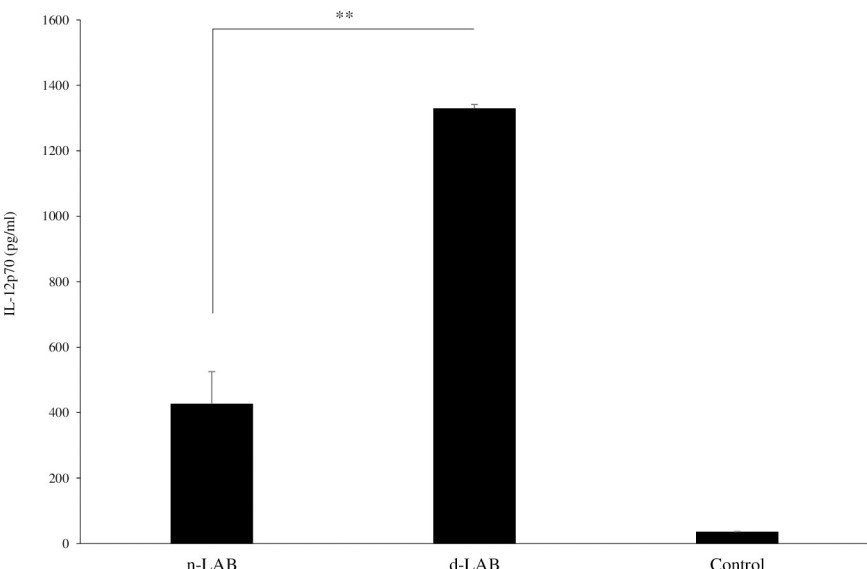

**Fig 4. Effect of n-LAB and d-LAB administration on IL-12 production in mouse splenocytes.** n-LAB and d-LAB were co-cultured with mouse splenocytes for 24 h. IL-12 concentration in the culture supernatant was measured using enzyme-linked immunosorbent assay. Control, culture medium only; d-LAB, dispersed lactic acid bacteria; n-LAB, non-treated lactic acid bacteria. Each value is presented as the mean ± SD; n = 6; [**]p < 0.01 vs. n-LAB.

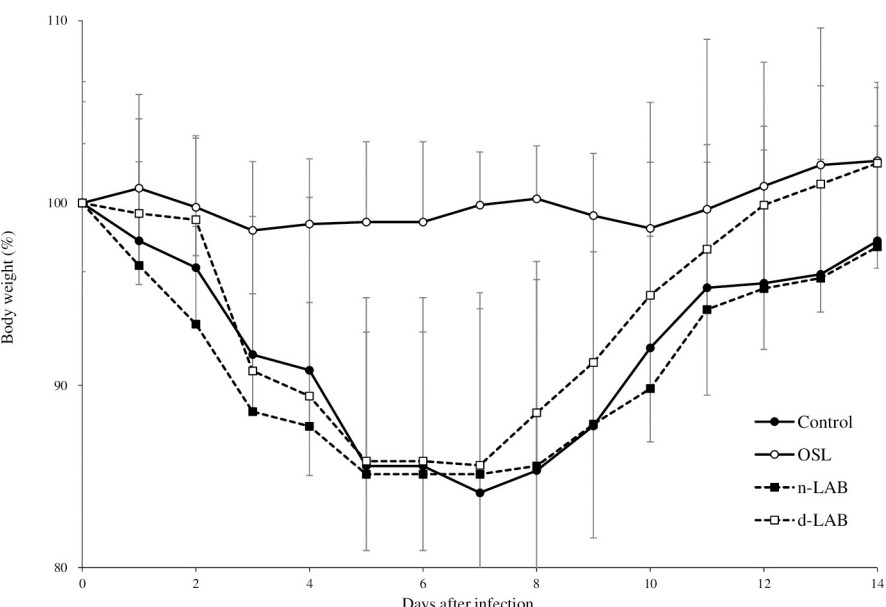

**Fig 5. Body weight change in mice infected with IFV.** IFV-infected mice were orally administered distilled water (control, filled circle), 0.2 mg/day of oseltamivir (OSL, white circle), 5 mg/day of non-treated lactic acid bacteria (n-LAB, filled square), and 5 mg/day of dispersed lactic acid bacteria (d-LAB, white square) from 7 day pre-infection to 14 day post-infection. Body weights are relative to those on the day of viral infection (day 0), which was set as 100%. Each value is presented as the mean ± SD; n = 5; IFV, influenza A virus.

Virus yields in the lungs and BALF of IFV-infected mice on day 3 post-infection are shown in Fig 6A and 6B, respectively. Oral n-LAB and d-LAB administration significantly reduced the viral load in the BALF (p < 0.01) and lungs (p < 0.05) compared with the control group.

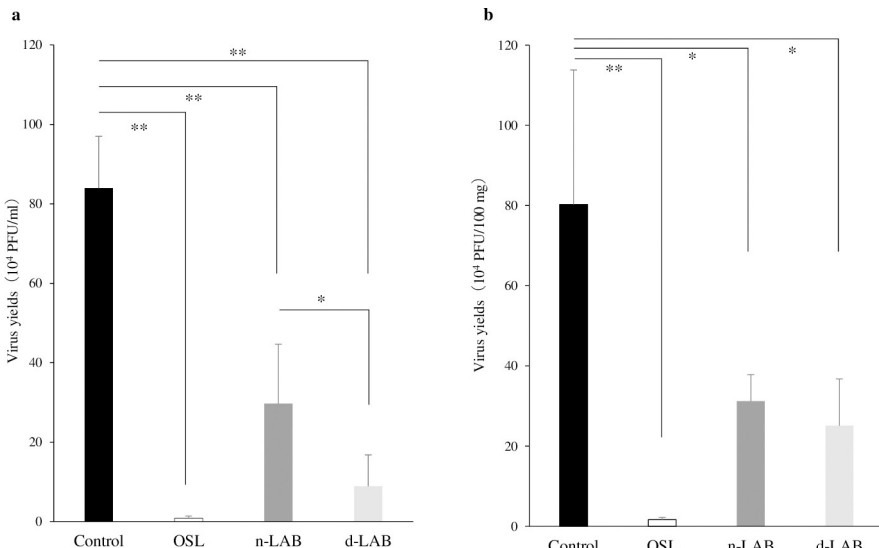

**Fig 6. Effect of LAB or d-LAB administration on viral load in mice.** Virus yield in BALF (**a**) and lung samples (**b**) were measured using a plaque assay on day 3 post-infection. Each value is presented as the mean ± SD; n = 5; \*\*p < 0.01; \*p < 0.05. BALF, bronchoalveolar lavage fluid; d-LAB, dispersed lactic acid bacteria; n-LAB, non-treated lactic acid bacteria; OSL, oseltamivir; PFU, plaque-forming unit.

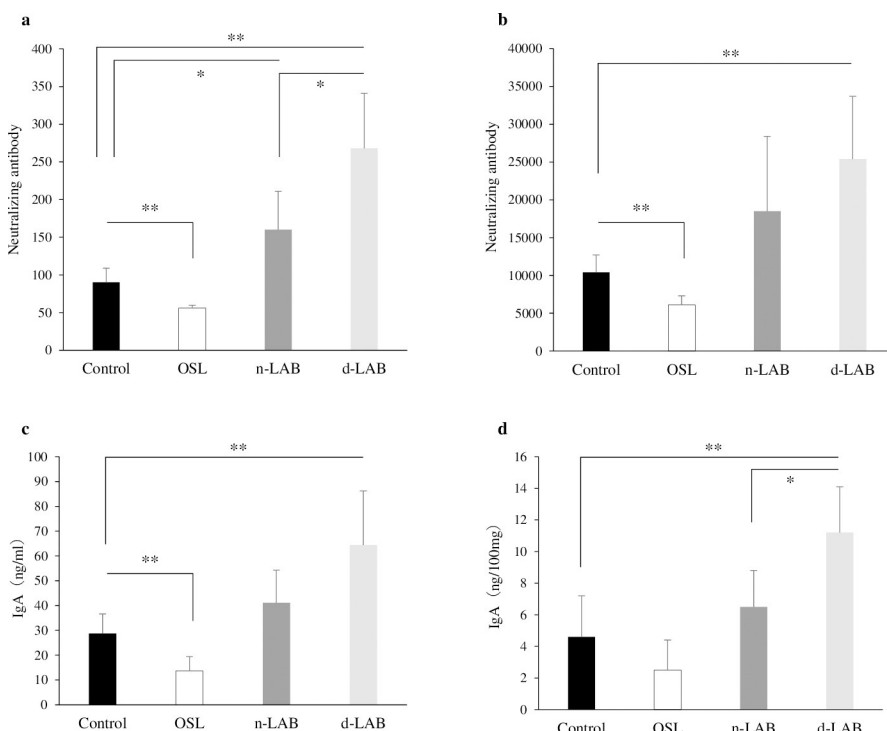

**Fig 7. Effects of n-LAB or d-LAB administration on the neutralizing antibody titer and IFV-specific IgA production in mice.** The titer of the virus-neutralizing antibody is presented as the reciprocal of the dilution of BALF samples (**a**) and sera (**b**) that reduced the plaque number to a level below 50% of that observed in the virus control. The IFV-specific IgA levels in BALF (**c**) and fecal samples (**d**) were determined using enzyme-linked immunosorbent assay. Each value is presented as the mean ± SD; n = 5; $^{**}$p < 0.01; $^{*}$p < 0.05. BALF, bronchoalveolar lavage fluid; IFV, influenza A virus; d-LAB, dispersed lactic acid bacteria; n-LAB, non-treated lactic acid bacteria; OSL, oseltamivir.

The d-LAB group showed a significantly lower viral load in the BALF samples than the n-LAB group (p < 0.05). Viral loads in the OSL group were markedly low, as shown in Fig 6A and 6B.

Fig 7A and 7B show the effects of n-LAB and d-LAB on the neutralizing antibody response to IFV in BALF samples (Fig 7A) and sera (Fig 7B) at day 14 post-infection. The antibody titers of BALF samples obtained from the mice in the n-LAB or d-LAB group were significantly higher than those in the control group. Moreover, it was significantly higher in the d-LAB group than that in the n-LAB group (p < 0.05). By contrast, the antibody titer in the d-LAB group was significantly higher (p < 0.01) than that in the control group.

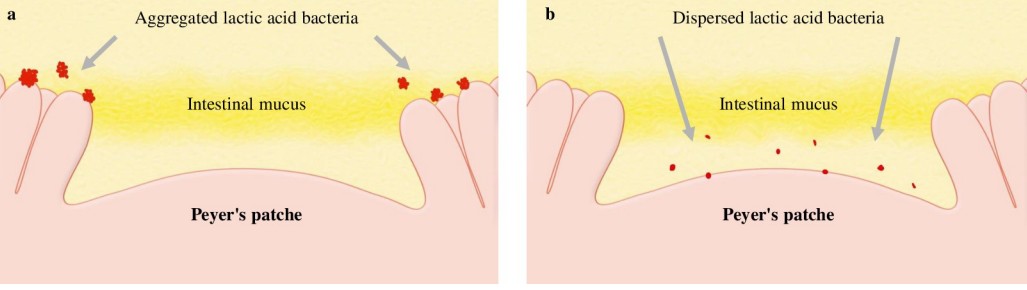

**Fig 8. Image of n-LAB or d-LAB in contact with the small intestinal Peyer's patches.** n-LAB could not reach the Peyer's patches because of large particle size, which is blocked by mucus (a). d-LAB were small particle size and can pass via mucus to reach the Peyer's patches (b). d-LAB, dispersed lactic acid bacteria; n-LAB, non-treated lactic acid bacteria.

To elucidate whether n-LAB and d-LAB stimulate the local immune response in mice, the levels of IFV-specific IgA in BALF and fecal samples were determined at day 14 post-infection (Fig 7C and 7D). IgA production in the d-LAB group was significantly higher than that in the control group (p < 0.01). The IgA levels in the feces were significantly higher in the d-LAB group compared with the n-LAB group (p < 0.05).

## Discussion

When cultured LAB were washed and dried into powder form, the bacteria adhered to each other and formed aggregates (Table 1, Fig 2). A possible cause for this phenomenon is that the polysaccharides produced by LAB and the capsules present on bacterial surface may help in bonding during drying. Furthermore, bacterial surface proteins are reportedly involved in charge stability and that non-specific electrostatic effects may be a factor in adhesion [34]. LAB were reported to adhere to polysaccharides, such as xylan and mucin, by surface layer protein action [35], and it is natural for bacteria to adhere to a variety of substrates for survival. Thus, polysaccharide-producing LAB may adhere to each other. Further, powderization may strengthen the binding. However, in consideration of quality and cost, it is desirable to powderize LAB for distribution; however, if LAB agglomerate with each other following powderization (Fig 2A), the product may be affected. Therefore, we developed a LAB powder (d-LAB) with less agglomerates, by homogenizing the culture and adding dextrin to the powder for preventing the agglomerates from re-agglomerating (Fig 2B). d-LAB was measured using a laser diffraction particle size analyzer and the mean particle size was determined to be 0.679 μm, indicating few bacterial aggregates. Thereafter, n-LAB, with several bacterial aggregates, and d-LAB, with excellent water dispersibility and fewer aggregates, were administered to mice and observed in the vicinity of the Peyer's patches in the small intestine. n-LAB failed to reach the Peyer's patches and drifted on the mucus covering these patches, whereas d-LAB reached the Peyer's patches. This difference may be owing to the presence of membrane-tethered mucin in the intestinal cells. [36, 37]. We believe that n-LAB, which contains several bacterial aggregates of approximately 50 μm in size (Fig 3A), is physically prevented from contacting the intestinal tract by mucin, whereas d-LAB, which has fewer aggregates, can pass through mucin and reach the Peyer's patches (Fig 8A and 8B). In the future, we intend to investigate whether the aggregation or size of the bacteria affects the passage via the mucin layer. Furthermore, larger LAB aggregates were found to reduce IL-12 production by phagocytosis of mouse splenocytes (Fig 4). These results suggest that LAB aggregation reduces their uptake from M cells in the Peyer's patches as well as affects their phagocytosis and decreases the immune response of LAB; therefore, we compared n-LAB and d-LAB in a mouse influenza infection model. n-LAB significantly differed from d-LAB in the IFV yield of BALF at 3 days post-infection (Fig 6) and in the neutralization of antibody titers of BALF samples and IFV-specific IgA in feces at 14 days post-infection (Fig 7). Although the difference was not significant, n-LAB showed almost the same transition in body weight as control, whereas d-LAB showed rapid weight recovery after 7 days of infection and returned to pre-infection weight by day 12 (Fig 5). The results of the present study showed that the effectiveness of LAB decreases when there are numerous LAB aggregates. In the future, we intend to confirm the influence of bacterial agglutination on health effects by comparing and verifying different methods of powdering and different species of bacteria. The effects of heat-killed LAB have extensively been researched [38–40]. However, reports of changes in the physical properties of LAB during powderization affecting their health benefits are scarce. To render LAB use effective, we recommend studying and verifying the properties of LAB at the consumer stage. Accordingly, we intend to evaluate the LAB prepared in our laboratory as well as study LAB in a form similar to

the final product for confirming whether the effectiveness of LAB has disappeared. We will continue to develop better LAB products and improve their quality.

## Conclusions

In the present study, it was found that LAB could not reach the Peyer's patches via the intestinal mucosa because of aggregate formation when LAB were powdered. LAB containing numerous aggregates was found to impact its efficacy (protection against influenza infection). Therefore, we developed a LAB (d-LAB) with a lower number of agglomerates by treating it with a high-pressure homogenizer and adding an excipient to prevent agglomeration. When d-LAB were administered to mice, they were able to pass via the intestinal mucosa and reach the Peyer's patches because of the improved water dispersibility. Furthermore, d-LAB was found to be more effective in mouse models of IFV infection. Therefore, we can increase the health benefits of LAB by improving the low water dispersibility of LAB.

## Acknowledgments

We thank Ms. Kazumi Shimizu (Non-Profit Organization, The Japanese Association of Clinical Research on Supplements) and Ms. Yuriko Namatame (Bio-Lab Co., Ltd.) for skillful technical assistance and valuable discussions. The authors would like to thank Enago (www.enago.jp) for the English language review.

## Author Contributions

**Conceptualization:** Takumi Watanabe.

**Data curation:** Kyoko Hayashi.

**Formal analysis:** Kyoko Hayashi.

**Funding acquisition:** Takumi Watanabe, Isao Takahashi, Tatsuhiko Kan.

**Investigation:** Takumi Watanabe, Kyoko Hayashi.

**Methodology:** Takumi Watanabe, Kyoko Hayashi, Isao Takahashi, Makoto Ohwaki.

**Project administration:** Toshio Kawahara.

**Resources:** Tatsuhiko Kan, Toshio Kawahara.

**Software:** Takumi Watanabe, Kyoko Hayashi, Makoto Ohwaki.

**Supervision:** Toshio Kawahara.

**Validation:** Kyoko Hayashi.

**Visualization:** Takumi Watanabe, Isao Takahashi.

**Writing – original draft:** Takumi Watanabe.

**Writing – review & editing:** Kyoko Hayashi, Tatsuhiko Kan, Toshio Kawahara.

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
