## [Decision Letter · Decision Letter 0]

31 Dec 2020

PONE-D-20-33627

Physical Properties of Lactic Acid Bacteria Influence the Level of Protection Against Influenza Infection in Mice

PLOS ONE

Dear Dr. Watanabe,

Thank you for submitting your manuscript to PLOS ONE. After careful consideration, we feel that it has merit but does not fully meet PLOS ONE’s publication criteria as it currently stands. Therefore, we invite you to submit a revised version of the manuscript that addresses the points raised during the review process.

We look forward to receiving your revised manuscript.

Kind regards,

Wenke Feng, PhD

Academic Editor

PLOS ONE

Journal Requirements:

2. Our staff editors have determined that your manuscript is likely within the scope of our Call for Papers on Influenza. This editorial initiative is headed by PLOS ONE Guest Editors Dr. Meagan Deming and Dr. Deshayne Fell. The Collection encompasses research on influenza prevention on every level, including in vitro, translational, behavioral, and clinical studies; disease and immunity modelling; as well as new approaches to influenza prevention. Additional information can be found on our announcement page: https://collections.plos.org/call-for-papers/influenza/.

Currently, your manuscript is included in the group of papers being considered for this call. Please note that being considered for the Collection does not require additional peer review beyond the journal’s standard process and will not delay the publication of your manuscript if it is accepted by PLOS ONE. We would greatly appreciate your confirmation that you would like your manuscript to be considered for this Collection by indicating this in your next cover letter. If you would prefer to remove your manuscript from collection consideration, please specify this in your cover letter.

We note that one or more of the authors are employed by a commercial company: Bio-Lab Co., Ltd. and ICAM Co., Ltd.

3.1. Please provide an amended Funding Statement declaring this commercial affiliation, as well as a statement regarding the Role of Funders in your study. If the funding organization did not play a role in the study design, data collection and analysis, decision to publish, or preparation of the manuscript and only provided financial support in the form of authors' salaries and/or research materials, please review your statements relating to the author contributions, and ensure you have specifically and accurately indicated the role(s) that these authors had in your study. You can update author roles in the Author Contributions section of the online submission form.

3.2. Please also provide an updated Competing Interests Statement declaring this commercial affiliation along with any other relevant declarations relating to employment, consultancy, patents, products in development, or marketed products, etc.  

3.3. Please know it is PLOS ONE policy for corresponding authors to declare, on behalf of all authors, all potential competing interests for the purposes of transparency. PLOS defines a competing interest as anything that interferes with, or could reasonably be perceived as interfering with, the full and objective presentation, peer review, editorial decision-making, or publication of research or non-research articles submitted to one of the journals. Competing interests can be financial or non-financial, professional, or personal. Competing interests can arise in relationship to an organization or another person. Please follow this link to our website for more details on competing interests: http://journals.plos.org/plosone/s/competing-interests

Reviewers' comments:

Reviewer's Responses to Questions

**Comments to the Author**

1. Is the manuscript technically sound, and do the data support the conclusions?

Reviewer #1: Partly

Reviewer #2: Partly

2. Has the statistical analysis been performed appropriately and rigorously? 

Reviewer #1: Yes

Reviewer #2: Yes

3. Have the authors made all data underlying the findings in their manuscript fully available?

Reviewer #1: Yes

Reviewer #2: Yes

4. Is the manuscript presented in an intelligible fashion and written in standard English?

Reviewer #1: Yes

Reviewer #2: Yes

5. Review Comments to the Author

Reviewer #1: The paper is of some interest and provides evidence that more effective dispersion of a killed lactic acid bacteria preparation is possible and that this leads to greater activity in an in vivo model vs influenza virus. The main positive findings include reduced viral load in BALF samples at day 3 (and to a lesser extent in whole lung samples) and increased antiviral IgA response compared with non dispersed bacteria. There is evidence of increased IL-2 production with the dispersed preparation from isolated spleen cells. The topic is of interest especially since this seems to be a potentially benign way of reducing severity of lung viral infection through oral ingestion of killed and powderized bacteria. The critiques are as follows:

1. The microscopy seems to be of poor quality and it is hard to assess in the images the location of the intestinal cells in the background. The confocal microscopy is somewhat better but it is necessary to show the results of the aggregated bacteria on confocal microscopy as well as the dispersed.

2. It should be elucidated what the basis for assuming that the bacterial mechanism of action mainly involved binding to Peyer's Patches.

3. It should be made clear in the discussion that the theory that the aggregated bacteria cannot traverse mucin layers in speculative.

4. The discussion also notes that there is different interaction of the dispersed bacteria with macrophages and dendritic cells but it is not clear from the data presented that these are the cells involved in generating the IL-12.

5. There should be some discussion of why the splenocyte and IL-12 experiments were done.

6. It should be made clear if the rate of recovery of weight was statistically different for the dispersed bacteria.

7. Figure 4 title refers to ribonuclease treatment - Where does the ribonuclease come in?

Reviewer #2: In this study, the authors investigated how the powderization of lactic acid bacteria (LAB) affects their water dispersibility and their efficacy in inducing immune response in a mouse model of IFV infection. Two different types of powdered Enterococcus faecalis KH2 preparation (non-treated and dispersed) were compared in their particle size, the distribution in the small intestine and the protective effects of these LAB preparations against viral infection in a mouse model of influenza infection. The authors showed that by treating LAB with a high-pressure homogenizer and adding dextrin in the preparation can dramatically decrease the formation of bacteria aggregates, which increased the water dispersibility of the bacteria and thus enhanced their uptake by intestinal Peyer’s patches and improved their protection against virus infection. The experiments were well-designed, and the conclusion was reasonably sound. However, there are several major concerns that should be addressed:

1. Several abbreviations were not defined in the manuscript and should be added in the main text where it first appeared (eg. PRRs, M cells, et al).

2. Line 62, “to investigate whether the effect of the difference in the LAB species or powderization was due to the difference in the species” This statement is really confusing and should be rephrased. Also, were the species of bacteria in the LAB powdered preparation different in n-LAB and d-LAB?

3. Typos and grammar issues should be double-checked and corrected.

4. In the evaluation of LAB’s distribution in mouse Peyer’s patches, male specific pathogen-free mice were used, whereas female mice were used in the influenza infection model. Are there any particular reasons in choosing female mice in the infection model? And have you looked at the bacteria distribution in Peyer’s patches in female mice as well? Is it different from male mice, can you comment on this?

5. The distribution of LAB in mouse intestine was evaluated in male mice injected with LAB and harvested after 1h in this manuscript, LAB preparations were also administered to female mice by gavaging in the IFV infection model. Have you looked at the distribution of LAB in mice with IFV infection? How does the distribution change overtime before and after virus infection? Can you comment on this?

6. Line 99, the LAB suspension was added “to six wells of a 96-well cell culture plate”, is this correct?

7. In the IFV infection model, n-LAB, d-LAB and OSL were given two times per day from 7 days before viral inoculation to 14 days after inoculation. Were there any differences in body weight change in the first 7 days with LAB administration before viral inoculation? Have you tried different treatment schemes (pre-treatment, LAB administration during infection, or LAB after infection) with LAB preparations in the infection model and were there any differences?

8. Figure legends should be placed following the figure or combined and placed after the main text of the manuscript.

9. Virus loads in BALFs and lungs were determined on day 3 after IFV infection and neutralizing antibody and IgA levels were determined on day 14. Have you tested the virus load in BALFs and lungs on day 3 as well?

10. In Figure 3, a and b seems showing same magnification based on the scale bar, however the images seem to be taken under obviously different magnification scale, this should be addressed and corrected, if any applicable. Additionally, the indications of different fluorescence colors presenting in Figure 3c should be added in the figure legend.

11. Line 181-184. This part of result was written poorly. The rationale and hypothesis of this specific experiments set should be addressed and a conclusion from the observations should be made or at least discussed.

12. Influenza infected mice treated with d-LAB lost weight at a slower rate and recovered faster compared to those treated with n-LAB, was the difference statistically significant? The signifiers should be added to the graphs if there is any.

13. Figure 6a, significance bar misplaced. Please correct this.

14. Line 207, The d-LAB group had a significantly lower viral load than n-LAB group in the BALFs but no differences were seen in the lung virus yields, can you explain this?

15. A control group of influenza infected mice treated with LAB culture preparation without powderization should be added, in comparison to different types of powdered LAB preparations.

16. Are there any other aspects other than the formation of aggregates that could possibly affect the beneficial effects of LAB? (eg, bacteria morphology changes during and after powderization, biofilm formation, et al.)

6. PLOS authors have the option to publish the peer review history of their article (what does this mean?). If published, this will include your full peer review and any attached files.

Reviewer #1: **Yes: **Kevan Hartshorn

Reviewer #2: No

---

## [Author Response · Author response to Decision Letter 0]

16 Apr 2021

Updated statements

Funding

The research received support from Bio-Lab Co., Ltd. and ICAM Co., Ltd. The funder provided support in the form of salaries for authors (Takumi Watanabe, Tatsuhiko Kan, and Isao Takahashi) but had no role in study design, data collection and analysis, decision to publish, or preparation of the manuscript.

Competing interests

Takumi Watanabe and Tatsuhiko Kan are employed by Bio-Lab Co., Ltd. Isao Takahashi is employed by ICAM Co., Ltd. None of the authors had a personal or financial conflict of interest. This does not alter the authors’ adherence to PLOS ONE policies on sharing data and materials. In addition, all authors involved with the present manuscript declare that they have no competing interests of any kind, including financial and non-financial competing interests.

Responses to Reviewer #1:

We thank the Reviewer for the insightful comments. We believe that the comments have helped us significantly improve the paper. In particular, we would like to express our gratitude for the opportunity to consider the mechanism of action of lactic acid bacteria in further detail.

1. The microscopy seems to be of poor quality and it is hard to assess in the images the location of the intestinal cells in the background. The confocal microscopy is somewhat better but it is necessary to show the results of the aggregated bacteria on confocal microscopy as well as the dispersed.

Response: An image from the lumen side was added to illustrate the relationship between intestinal cells and bacteria (Fig 3c). Because n-LAB (aggregated bacteria) cannot be observed on epithelial cells of Peyer’s patches, we did not detect them using confocal microscopy. As mentioned by you, a comparison between aggregated and dispersed bacteria should be performed; accordingly, we intend to collect the Peyer’s patches and count the number of bacteria in future research.

2. It should be elucidated what the basis for assuming that the bacterial mechanism of action mainly involved binding to Peyer's Patches.

Response: Thank you for your important comment. We have added that bacterial S-layer protein is absorbed via binding to uromodulin of M cells in Peyer’s patches [L57-59]. A reference of previous studies [14] has been added.

3. It should be made clear in the discussion that the theory that the aggregated bacteria cannot traverse mucin layers in speculative.

Response: As mentioned by you, it is a speculation; accordingly, we intend to conduct research on the mucus on the Peyer's patches to determine if there is a difference in passage of particles owing to aggregation. We have added content to clarify this [L283-284].

4. The discussion also notes that there is different interaction of the dispersed bacteria with macrophages and dendritic cells but it is not clear from the data presented that these are the cells involved in generating the IL-12.

Response: The description of the cells used (mouse splenocytes) was incorrect and has been rectified [L286-287].

5. There should be some discussion of why the splenocyte and IL-12 experiments were done.

Response: Thank you for this comment. To clarify this, in the Introduction, we have added a sentence about verifying the effect of bacterial aggregation on IL-12 production using mouse splenocytes and two appropriate references [L77-78].

6. It should be made clear if the rate of recovery of weight was statistically different for the dispersed bacteria.

Response: There was no significant difference in weight recovery of mice in the d-LAB group when compared with other groups or during the 7-day administration period. We have added sentences to clarify this [L221-225].

7. Figure 4 title refers to ribonuclease treatment - Where does the ribonuclease come in?

Response: We apologize for the error and have rectified it [L209].

Responses to Reviewer #2:

We wish to express our appreciation to the Reviewer for the insightful comments, which have helped us significantly improve the paper. We also thank you for your valuable suggestions for future research.

1. Several abbreviations were not defined in the manuscript and should be added in the main text where it first appeared (eg. PRRs, M cells, et al).

Response: Thanks for bringing this to our notice. We have rectified these instances (PRRs, M cell, and DAPI).

2. Line 62, “to investigate whether the effect of the difference in the LAB species or powderization was due to the difference in the species” This statement is really confusing and should be rephrased. Also, were the species of bacteria in the LAB powdered preparation different in n-LAB and d-LAB?

Response: We agree that the context was difficult to understand and apologize for the same. We have revised it [L66-71]. n-LAB and d-LAB are the same strain of bacteria.

3. Typos and grammar issues should be double-checked and corrected.

Response: We asked a private proofreading company (Enago) to re-check the manuscript and rectify such errors.

4. In the evaluation of LAB’s distribution in mouse Peyer’s patches, male specific pathogen-free mice were used, whereas female mice were used in the influenza infection model. Are there any particular reasons in choosing female mice in the infection model? And have you looked at the bacteria distribution in Peyer’s patches in female mice as well? Is it different from male mice, can you comment on this?

Response: Female mice were used in these experiments because they have been used in our past influenza infection studies and other studies. One of the reasons for this is that female mice have a calmer temperament, which reduces the stress on the mice and the experimenter during continuous administration. We have not assessed bacterial distribution in female mice in the present study; however, we have previously confirmed that they were absorbed by the Peyer’s patches in the other experiments using the same species of BALB/c mice. However, as mentioned by you, there may be differences between males and females; accordingly, we intend to clarify the differences between males and females in future studies.

5. The distribution of LAB in mouse intestine was evaluated in male mice injected with LAB and harvested after 1h in this manuscript, LAB preparations were also administered to female mice by gavaging in the IFV infection model. Have you looked at the distribution of LAB in mice with IFV infection? How does the distribution change overtime before and after virus infection? Can you comment on this?

Response: We have not observed any distribution of LAB in IFV-infected mice. As mentioned in your comment, IFV infection may have changed the distribution; thus, we would like to confirm the effect of infection and distribution in future studies.

6. Line 99, the LAB suspension was added “to six wells of a 96-well cell culture plate”, is this correct?

Response: We apologize for the confusing notation. It has been corrected to “6 wells per sample” [L109].

7. In the IFV infection model, n-LAB, d-LAB and OSL were given two times per day from 7 days before viral inoculation to 14 days after inoculation. Were there any differences in body weight change in the first 7 days with LAB administration before viral inoculation? Have you tried different treatment schemes (pre-treatment, LAB administration during infection, or LAB after infection) with LAB preparations in the infection model and were there any differences?

Response: There were no differences in body weight change in each group during the 7 days of administration pre-infection. This information has been added to the text [L224-225]. The comparison of lactic acid bacteria in different treatment schemes is of interest to us and we will continue our research to further clarify these points.

8. Figure legends should be placed following the figure or combined and placed after the main text of the manuscript.

Response: The PLOS ONE submission guidelines instructed that the figure legends should be provided separately from the figures. As per the guidelines, figures should not be included in the main manuscript file. Accordingly, we have provided separate figures in the formats specified by the guidelines.

9. Virus loads in BALFs and lungs were determined on day 3 after IFV infection and neutralizing antibody and IgA levels were determined on day 14. Have you tested the virus load in BALFs and lungs on day 3 as well?

Response: The comment was slightly ambiguous. If your query is regarding virus detection on day 14 of infection, no virus was detected on day 14 in the all groups. The virus was not detected in both BALF and lung samples by day 7 after virus inoculation.

10. In Figure 3, a and b seems showing same magnification based on the scale bar, however the images seem to be taken under obviously different magnification scale, this should be addressed and corrected, if any applicable. Additionally, the indications of different fluorescence colors presenting in Figure 3c should be added in the figure legend. 

Response: The images (a) and (b) in Fig 3 are of the same magnification. However, n-LAB in (a) appears to be larger because of the agglomeration; thus, the magnification may be perceived differently. Therefore, in the Results section, we have added that Fig.3a shows a larger bacterial image than Fig.3b owing to bacterial aggregation [L190-191]. Different fluorescent color indications have been added to the figure legend.

11. Line 181-184. This part of result was written poorly. The rationale and hypothesis of this specific experiments set should be addressed and a conclusion from the observations should be made or at least discussed.

Response: We agree that we needed to clarify the rationale of the experiment. We have made corrections and additions in the Introduction and Discussion sections for clarity [L77-78, L286-287].

12. Influenza infected mice treated with d-LAB lost weight at a slower rate and recovered faster compared to those treated with n-LAB, was the difference statistically significant? The signifiers should be added to the graphs if there is any.

Response: No significant difference was found. However, because it was an interesting pattern, we have added the number of days required to return to pre-infection weight for the Control, n-LAB, and d-LAB groups [L221-224].

13. Figure 6a, significance bar misplaced. Please correct this.

Response: Thanks for bringing this to our notice. We have rectified it.

14. Line 207, The d-LAB group had a significantly lower viral load than n-LAB group in the BALFs but no differences were seen in the lung virus yields, can you explain this?

Response: There was a substantial variation between individual mice of the d-LAB group in terms of the measured virus yields. We presumed that this variation is the cause for the difference observed.

15. A control group of influenza infected mice treated with LAB culture preparation without powderization should be added, in comparison to different types of powdered LAB preparations.

Response: Thank you for your comments. We agree and consider it to be important as well. However, because this study was conducted with limited resources, we would like to compare and verify the results with LAB culture preparation without powderization and with LAB powderized using different methods in our ongoing research. We added an explanation about this in the Discussion [L296-297].

16. Are there any other aspects other than the formation of aggregates that could possibly affect the beneficial effects of LAB? (eg, bacteria morphology changes during and after powderization, biofilm formation, et al.)

Response: Thank you for your targeted question. We believe that heat sterilization temperature may be an important factor. We are currently preparing heat-killed lactic acid bacteria with different sterilization temperatures and studying the immune response in vitro and in vivo. We are confirming the difference depending on the sterilization temperature and will report the results at the earliest.

---

## [Decision Letter · Decision Letter 1]

4 May 2021

Physical properties of lactic acid bacteria influence the level of protection against influenza infection in mice

PONE-D-20-33627R1

Dear Dr. Watanabe

We’re pleased to inform you that your manuscript has been judged scientifically suitable for publication and will be formally accepted for publication once it meets all outstanding technical requirements.

Kind regards,

Wenke Feng, PhD

Academic Editor

PLOS ONE

Additional Editor Comments (optional):

Reviewers' comments:

Reviewer's Responses to Questions

**Comments to the Author**

1. If the authors have adequately addressed your comments raised in a previous round of review and you feel that this manuscript is now acceptable for publication, you may indicate that here to bypass the “Comments to the Author” section, enter your conflict of interest statement in the “Confidential to Editor” section, and submit your "Accept" recommendation.

Reviewer #1: All comments have been addressed

Reviewer #2: All comments have been addressed

2. Is the manuscript technically sound, and do the data support the conclusions?

Reviewer #1: Yes

Reviewer #2: Yes

3. Has the statistical analysis been performed appropriately and rigorously? 

Reviewer #1: Yes

Reviewer #2: Yes

4. Have the authors made all data underlying the findings in their manuscript fully available?

Reviewer #1: Yes

Reviewer #2: Yes

5. Is the manuscript presented in an intelligible fashion and written in standard English?

Reviewer #1: Yes

Reviewer #2: Yes

6. Review Comments to the Author

Reviewer #1: (No Response)

Reviewer #2: The authors have addressed all the comments and questions in a comprehensive way, the suggested information were also added to the main text.

7. PLOS authors have the option to publish the peer review history of their article (what does this mean?). If published, this will include your full peer review and any attached files.

Reviewer #1: **Yes: **Kevan Hartshorn

Reviewer #2: No

---

## [Editor Report · Acceptance letter]

7 May 2021

PONE-D-20-33627R1 

Physical properties of lactic acid bacteria influence the level of protection against influenza infection in mice 

Dear Dr. Watanabe:

I'm pleased to inform you that your manuscript has been deemed suitable for publication in PLOS ONE. Congratulations! Your manuscript is now with our production department. 

Kind regards, 

on behalf of

Dr. Wenke Feng 

Academic Editor

PLOS ONE